# Bioacoustic Detection of Wolves: Identifying Subspecies and Individuals by Howls

**DOI:** 10.3390/ani12050631

**Published:** 2022-03-02

**Authors:** Hanne Lyngholm Larsen, Cino Pertoldi, Niels Madsen, Ettore Randi, Astrid Vik Stronen, Holly Root-Gutteridge, Sussie Pagh

**Affiliations:** 1Department of Chemistry and Bioscience, Aalborg University, 9220 Aalborg, Denmark; cp@bio.aau.dk (C.P.); nm@bio.aau.dk (N.M.); erandi@bio.aau.dk (E.R.); astrid.stronen@gmail.com (A.V.S.); sup@bio.aau.dk (S.P.); 2Department of Biology, Biotechnical Faculty, University of Ljubljana, 1000 Ljubljana, Slovenia; 3Animal Behaviour, Cognition and Welfare Group, University of Lincoln, Lincoln LN6 7TS, UK; hrootgutteridge@lincoln.ac.uk; 4School of Animal Rural and Environmental Sciences, Nottingham Trent University, Southwell NG25 0QF, UK

**Keywords:** bioacoustics, *Canis lupus*, discriminant analysis, habitats directive, monitoring, fundamental frequency, acoustic variables

## Abstract

**Simple Summary:**

This study evaluates the use of acoustic devices as a method to monitor wolves by analyzing different variables extracted from wolf howls. By analyzing the wolf howls, we focused on identifying individual wolves, subspecies. We analyzed 170 howls from 16 individuals from the three subspecies: Arctic wolves (*Canis lupus arctos*), Eurasian wolves (*C.l. lupus*), and Northwestern wolves (*C.l. occidentalis*). We assessed the potential for individual recognition and recognition of three subspecies: Arctic, Eurasian, and Northwestern wolves.

**Abstract:**

Wolves (*Canis lupus*) are generally monitored by visual observations, camera traps, and DNA traces. In this study, we evaluated acoustic monitoring of wolf howls as a method for monitoring wolves, which may permit detection of wolves across longer distances than that permitted by camera traps. We analyzed acoustic data of wolves’ howls collected from both wild and captive ones. The analysis focused on individual and subspecies recognition. Furthermore, we aimed to determine the usefulness of acoustic monitoring in the field given the limited data for Eurasian wolves. We analyzed 170 howls from 16 individual wolves from 3 subspecies: Arctic *(Canis lupus arctos*), Eurasian (*C. l. lupus*), and Northwestern wolves (*C. l. occidentalis*). Variables from the fundamental frequency (f0) (lowest frequency band of a sound signal) were extracted and used in discriminant analysis, classification matrix, and pairwise post-hoc Hotelling test. The results indicated that Arctic and Eurasian wolves had subspecies identifiable calls, while Northwestern wolves did not, though this sample size was small. Identification on an individual level was successful for all subspecies. Individuals were correctly classified with 80%–100% accuracy, using discriminant function analysis. Our findings suggest acoustic monitoring could be a valuable and cost-effective tool that complements camera traps, by improving long-distance detection of wolves.

## 1. Introduction

In 2012, a wolf (*Canis lupus lupus*) was found dead in northern Jutland, Denmark, which was the first observation of wolves in Denmark since 1813 [1]. The wolves in Denmark are dispersers from Germany and their descendants, and are part of a connected Central European wolf population. In Europe (excluding Russia), there are more than 17,000 wolves [2]. In the European Union (EU), the wolf population is estimated to exceed 13,000 individuals [2] and the European populations are generally increasing in size due to recent protection. However, in most Western European countries the populations are still relatively small with less than 1000 wolves [1,2]. In Scandinavia, the population is approximately 480 wolves [3], and there are at least a further 780 individuals found in Germany and western Poland [2]. Wolves dispersing from the Central European population have reached Belgium, the Netherlands, and Denmark [1,2]. The number of wolves present in Denmark was estimated as a litter of 4 pups and 11 adult individuals in the period 1 April to 30 June 2021; in total, it was estimated that there were eight immigrant adults and seven Danish born individuals [4]. However, monitoring the population and dispersal of individuals has proved to be challenging as wolves are both wide-ranging [5,6] and notoriously fearful of humans [7].

As in the rest of Europe, Denmark is obligated to monitor wolves according to the EU Habitats Directive (92/43/EEC) [8]. In Scandinavia and Germany, databases containing validated wolf observations, reported by citizens and by volunteers, and DNA samples obtained from scats and the carcasses of killed domestic animals have been established to track the population status of wolves [9,10,11]. A genetic reference database for wolves in Central Europe has been established to analyze the origin and movements of wolves across Europe [12].

Currently there is both active and passive monitoring of wolves in Denmark [1,13]. Active monitoring consists of registering tracks and DNA profiles, and the use of camera traps [14,15]. Passive monitoring is the registration of public observations, including organized volunteers as part of citizen science projects [14,15]. Camera traps are an essential tool but are limited by their small field of view compared to wolves’ extensive physical range, and their effective deployment requires prior knowledge of wolf presence in an area.

Acoustic monitoring is a passive monitoring tool that has been used in the last decades for studies of diverse taxa, including insects [16], bats [17,18], birds [19], whales [20,21], and large terrestrial mammals such as ungulates [22,23] and elephants [24], and with acoustic devices; it is similarly possible to detect elusive but vocal species such as wolves [25]. Recognizing individual wolves on their howls can give insight into their movements and territory size. Several studies have shown that acoustic monitoring of wolves can be a useful and relevant tool since it is cost-effective and non-invasive [26,27,28,29,30] and can help recognize wolves from a distance and determine the number of wolves present [25,28,30,31,32].

Acoustic monitoring of wolves is based in detection of their howls [26,33,34]. Wolves use howls for different purposes: (1) to defend their territory [26,33,35], (2) to contact members of the pack [26,35], and (3) to socialize [33,35]. They may howl solo or in chorus from packs [26,36]. Wolves usually howl around twilight and in the middle of the night [35] and their emissions are most intense during the months of July through October, when the presence of pups increases the pack’s howling [33]. This is likely to make other predators aware that the pups are protected by adults [33].

The fundamental frequency (f0) is the lowest frequency band of the tonal call as wolf howl. For wolves, the f0 typically has values between 150 and 780 Hz [37] and has been used to identify individual wolves with high accuracy within subspecies of wolves such as Eastern wolves (*Canis lupus lycaon*) [27,38], Indian wolves (*C.l. pallipes*) [39], and Iberian wolves (*C.l. signatus*) [36]. Root-Gutteridge et al. [40] were able to identify individuals in captive Eastern wolves with 100% accuracy using both f0 and amplitude (the highest variation of a wave in air pressure, perceived as volume). The use of amplitude for encoding individual identity for wolves in situ requires additional studies to determine the rate of sound suppression over distance, in different weather conditions, and through different habitats such as open land or forest [27]. Furthermore, wolves have been shown to have different vocal signatures depending on the subspecies [32,41] and are group specific [42].

Monitoring based on camera traps requires individuals in near field of view, which are most efficient at 10 m [43], whereas acoustic recorders cover larger distances with a with a radius of 3 km [28]. However, development of the methodology for analyzing the acoustic data is essential for wider scale use. Because of the small population in Denmark, captive wolves located in zoos are of importance to train the algorithm to recognize wolf howls. We know numbers present and can recognize individuals in zoos. Additionally, subspecies are known to show variations in their howls [41]; thus, recognition should be trained on the relevant subspecies.

The primary aim of this study is to analyze acoustic data on wolves’ howls collected from wild wolves and captive wolves from three subspecies Arctic (*C.l. arctos*), Eurasian (*C.l. lupus*), and Northwestern (*C.l. occidentalis*) wolves, including howls from both adults and infants. The analysis is focused on individual recognition and subspecies recognition. We aim to determine whether Eurasian wolf howls have the same encoding of individual identity and, thus, can potentially be used in individual tracking via bioacoustics monitoring. Furthermore, we aim to discuss the usefulness of acoustic monitoring in the field as an additional monitoring tool of wolves.

## 2. Materials and Methods

### 2.1. Collection of Acoustic Data

Data collections took place between July 2011 and April 2021 at one privately owned location in Toropets, Russia, and the UK Wolf Conservation Trust, and five different locations in Denmark. In Denmark, recordings were made in captivity at Ree Park Safari, which had seven Arctic wolves (three adults and four pups); Skandinavisk Dyrepark, which had three adult Eurasian wolves; and Givskud Zoo with seven adult Northwestern wolves. Recordings were also made in two in situ locations in Central Jutland in areas of mixed moor and forest (Table 1), known to be wolf territories. Furthermore, 47 recordings of captive Eurasian wolves were extracted from the Macaulay Sound Library and British Library Sound Archive with their permission. Full details are given in Table 1.

A Song Meter SM4 acoustic recorder (Wildlife Acoustics Inc., Maynard, MA, USA) was used for recordings in Denmark. The acoustic recorder had a sampling rate (digit capacity of samples per seconds of an audio recording) of 44.1 kHz and an amplitude resolution of 16 bits (digital capacity of amplitude values possible to record for each sample). The recorder was placed on the fences toward the wolves’ enclosure in the zoos and on trees in the wild in Central Jutland 1.5–2 m above ground. The recorder was set to auto record in 1–4 weeks from dusk till dawn (17:00–07:30) in a two-hour interval since this is the time the wolves are most active [35], and was saved in SD cards with 128 GB storage. In Givskud Zoo a portable sound amplifier Joyo JPA-863 (Joyo Technology Co., Ltd., Baoan, Shenzhen, China) was used to elicit howling. The howls were initiated by playing sounds for the wolves. Once they started howling the sound was stopped. When the wolves had calmed down and they no longer barked, howled, or whined another sound was played for them. This was continued for an hour with alternating sounds of ambulance sirens, church bells, and howls from a different wolf pack.

Howls obtained from the UK Wolf Trust were recorded with a UHER 4000 REPORT L recorder (UHER Werke, Munich, Germany). Wolves were videoed during all howling sessions and visually identified. Recordings were digitized to 44.1 kHz and 16 bits by the UK Wolf Trust.

Howls of a wild born but captive-held wolf were recorded in Russia by Dr. Holly Root-Gutteridge in 2011. Howls were elicited with both recordings played through a laptop and loudspeaker using three howls sampled from the file “Lonesome” recorded by Dr. Fred Harrington and by live howling. The response howls were recorded using a Sennheiser K6-ME67 (Sennheiser electronic GmbH & Co. KG, Wedemark, NI, Germany) directional long-range microphone, Light Snake USB (Soundtech, Milford, CT, USA) connecting cable, HP laptop (HP Inc., Palo Alto, CA, USA), M-Audio Microtrack portable recorder (Avisoft Bioacoustics e.K., Glienicke/Nordbahn, BB, Germany), and USB speakers to a sampling rate of 96 kHz and 24 bits. Where possible video was also recorded using a Sanyo Xacti CG20 digital video recorder (Sanyo, Osaka, Japan).

Howls obtained from British Library Sound Archive were recorded by Dr. Erik Zimen in the 1970s [44], as reported in the notes, and were digitized by the archive to a sampling rate of 44.1 kHz and 16 bits.

Howls obtained from Macaulay Sound Archive were recorded with UHER 4000 REPORT L recorder using a Sennheiser MKH104 condenser microphone and was digitized by Macaulay Sound Archive to a sampling rate of 44.1 kHz and 16 bits reported in the notes. The recordings were digitized by the archive.

All recordings were saved in the file format wav.

Fundamental frequency (f0) is the rate of which periodic sounds repeat itself [45] and are defined as the rate of which vocal folds vibrate [46]. By measuring the f0 variables, it is possible to characterize the sound and make comparisons between different vocalizations.

### 2.2. Call Analysis

The software Kaleidoscope pro 5.0.0 (Wildlife Acoustics Inc., Maynard, MA, USA) was used to train algorithms to identify wolf howls in the sound files. The signal parameters were set to 100–2000 Hz, 0.5–17 s in duration and the maximum inter-syllables gap (determines the end of a signal) was set to 0.35. For cluster analysis, the maximum distance from cluster center to include in output files (vocalizations detected are analyzed to determine if their similar) was set to 1.0. The maximum fast Fourier transformation (fft) (size influence the resolution of frequency over time) of 21.33 milliseconds (ms) was selected, with a maximum number of states (Hidden Markov models target size) at 12. The maximum distance to cluster center for building clusters (controls how cluster are formed from detected signals) was selected at 0.5. Maximum clusters (limits number of clusters formed) were set at 500. These settings were default for clustering except for the fft that was set at maximum. The detection was reviewed both by spectrogram and audio to manually assign the howls and the data were rescanned to train the software. Additional manual filtering was performed to remove cow and crow calls.

For the data collected in Denmark, Arctic, and Eurasian wolves were identified on their solo howls. Some of the wolves would howl unaccompanied for several minutes and were identified as one individual. The howls from the identified individual were compared to other single howls from the same location. Comparing spectrograms and sound it was possible to detect a difference in the fundamental frequency where one wolf from Ree Park Safari was consistently between 100 and 200 Hz lower than the other especially at the start of a howl. One wolf from Skandinavisk Dyrepark had a very distinct low start in their howls and made it possible to recognize. Howling Northwestern wolves were filmed and individually identified.

It was often impossible to extract the f0 of a wolf howl in choruses as several howls from different wolves would overlap; thus, only solo howls were used for the analysis. Recordings of howls from BLS028 and BLS029 were down sampled to 44.1 kHz and 16 bits in Audacity (Audacity Team^®^, USA) [47] before extracting the fundamental frequencies.

Sixteen acoustic variables (Table 2) were extracted from the howls using Praat (Praat, Amsterdam, Netherlands) [48] and a customized script in MATLAB (Mathworks Inc., Nattick, MA, USA) [49] developed previously [40], where 12 variables have been used in other studies [27,32,36,40]. The vocal parameters were measured by extracting the fundamental frequency (f0) contour of the calls using a cross-correlation method (Sound: To Pitch (cc) time step of 0.005 s, pitch floor 75 Hz, pitch ceiling 1200 Hz) [50]. In this study variables extracted from f0 were used for analysis. To make sure the f0 contour was tracked accurately, the contour of f0 was compared to the spectrogram (Figure 1) and if necessary, tracks of background noise such as wind or bird calls was removed. Some recordings had too much background noise, making it impossible for Praat to find the f0 and these recordings were excluded from the analysis.

### 2.3. Processing the Data

The howls recorded in Denmark were recorded in 2020–2021. During this study period, a total of 265 solo howls were recorded. Of these howls, 89 had too much wind noise to have accurate extractions and were removed. Of the remaining 176 howls collected, only 55 howls could be assigned to individual wolves (Table 1) were retained for further analysis. In Ree Park Safari and Skandinavisk Dyrepark we identified two out of three wolves. In Givskud Zoo we identified three out of seven adult wolves. Each of the seven wolves were named after the location and in order of the onset of howling, Wolf1, Wolf2, and Wolf3 (Table 1). Further, we used 50 wolf howls collected from The UK Wolf Conservation Trust and 18 Eurasian wolf howls collected in Russia by HRG. We also used archival howls: a further 41 Eurasian wolf howls from British Library Sound Archive, 6 Eurasian wolf howls from Macaulay Sound Library, The Eurasian wolves were given the letters BLS to show they were all from the same subspecies and not collected for this study.

Each wolf had their own ID number (BLS004 had the ID number 004 etc.) The Arctic wolves were called WCT with F for female and M for male. The first three letters were used to define them as being from the same source as well as the same subspecies.

### 2.4. Identification of Subspecies

Tests for differences in the acoustic variables were made with a one-way ANOVA and any variables with non-significant *p*-values (*p* > 0.05) were removed from further testing. Thus, the variables Minf0 and Cofv were non-significant (F_Minf0_ = 5.3, *p* > 0.05; F_Cofv_ = 2.5, *p* > 0.05) and were excluded from further tests. To avoid multicollinearity, we tested for correlation (Pearson’s correlation r > 0.6). Meanf0 were removed as they were highly correlated with Maxf0 (r = 0.91) and Endf0 (r = 0.87). Range was removed as it was highly correlated with Maxf0 (r = 0.74) and standard deviation (Sd) (r = 0.87). Posminf0 was highly correlated with Posmaxf0 (r = 0.59) and was removed.

A discriminant function analysis, hereafter discriminant analysis, was run in RStudio (RStudio, Boston, MA, USA) [52] and was applied to nine of the 16 acoustic variables (Maxf0, Posmaxf0, Sd, Slope, Cofm, IQR, Startf0, Endf0, and Duration) that were also used in studies discriminating among subspecies [32]. The discriminant analysis was applied to determine whether it is possible to correctly classify the already identified subspecies howls using these nine variables. A classification matrix was obtained to compare known howls to predicted howls. A pairwise post-hoc Hotelling test was used on the acoustic variables to test if there was a statistically significant difference between the howls in subspecies. In RStudio the datasets were randomly divided into smaller subsets making sure the number of howls were equal between all the subspecies. We ensured all the data from the large dataset were represented at least once in the subsets. This was performed as samples from different subspecies were not equal. A Sequential Bonferroni correction was applied to the pairwise post-hoc Hotelling test [54].

### 2.5. Individual Identification of Wolves from Howls

Northwestern wolves were excluded from the individual analysis as it was only possible to isolate two howls from two of the three wolves.

The same procedure for identifying subspecies was applied for individual identification. For Arctic wolves all variables were significant in the ANOVA. Pearson correlation showed that Meanf0 was highly correlated with Minf0 (r = 0.74), Maxf0 (r = 0.91) and Endf0 (r = 0.83) and was removed. Posminf0 was highly correlated with Posmaxf0 (r = 0.85) and was removed. Cofv was removed to ensure same variables in the analysis of both Eurasian and Arctic wolves. Slope was removed as it was highly correlated with Endf0 (r = 0.63). Range was removed as it was highly correlated with Maxf0 (r = 0.71) and Sd (r = 0.9). IQR was highly correlated with Startf0 (r = 0.69) and was removed (Table 3).

For Eurasian wolves all variables were significant with ANOVA. Pearson correlation showed that Meanf0 was highly correlated with Minf0 (r = 0.9), Maxf0 (r = 0.89), and Endf0 (r = 0.89) and was removed. Posminf0 was removed as it was highly correlated with Minf0 (r = 0.59). IQR was highly correlated with Minf0 (r = 0.84) and was removed. Cofv was highly correlated with Maxf0 (r = 0.72) and was removed. Slope was removed to ensure same variables in both analyses. Range was highly correlated with Endf0 (r = 0.71) and Startf0 (r = 0.62) and was removed (Table 3).

A discriminant analysis for was applied to eight of the 16 variables for the Arctic and Eurasian wolves (Minf0, Maxf0, Posmaxf0, Sd, Cofm, Startf0, Endf0, and Duration).

The discriminant analysis was applied to test if it was possible to correctly classify individual wolves from howls of already identified wolves within the same subspecies. A classification matrix was created to compare known howls to predicted howls. A pairwise post-hoc Hotelling test was run to test if there was a difference between the howls of individual wolves within the same subspecies. The same procedure of random subsets and sequential Bonferroni correction was applied to the identification of individual wolves.

## 3. Results

### 3.1. Identification of Subspecies

All the 170 howls from the seven wolves were used in the analysis for subspecies. The discriminant analysis using nine acoustic variables (Maxf0, Posmaxf0, Sd, Slope, Cofm, IQR, Startf0, Endf0, and Duration) gave an overall 78% of correctly classified subspecies (Figure 2). For Northwestern wolves the classification was 64%, for Arctic wolves, the classification was 82%, and finally for Eurasian wolves it was 77%. Pairwise post-hoc Hotelling test analysis showed significant difference in howls between Arctic and Eurasian wolves (DF = 9, F = 23.77, *p* < 0.001), Arctic and Northwestern wolves (DF = 9, F = 2.9, *p* < 0.01), and Eurasian and Northwestern wolves (DF = 9, F = 10, *p* < 0.001) all *p*-values were significant after sequential Bonferroni correction (Table 4). The randomized subsets gave overall correct classifications between 70% and 90% where most were between 82% and 88%. Pairwise post-hoc Hotelling test showed significant differences between most randomized subsets. After sequential Bonferroni 72% was not significant.

### 3.2. Individual Identification of Arctic Wolves

Sixty-two howls from four wolves were used in the analysis of howls from Arctic wolves. The discriminant analysis using eight variables (Minf0, Maxf0, Posmaxf0, Sd, Cofm, Startf0, Endf0, and Duration) gave an overall 95% correct classification of howls from the four wolves (Figure 3). Classification of each wolf showed that RW1, RW2 and WCTM all have 100% correct classification and WCTF has a 91% correct classification. The pairwise post-hoc Hotelling test showed significant difference in howls between RW1 and RW2 (DF = 8, F = 33.2, *p* < 0.001), between howls from RW1 and WCTM (DF = 8, F = 37.52, *p* < 0.001), between RW2 and WCTM (DF = 8, F = 9.68, *p* < 0.01), between RW1 and WCTF (DF = 8, F = 93,87, *p* < 0.001), between RW2 and WCTF (DF = 8, F = 41.74, *p* < 0.001) and lastly between WCTM and WCTF (DF = 8, F = 8.2, *p* < 0.01). All pairwise post-hoc Hotelling tests showed significant differences after sequential Bonferroni were applied (Table 5). After randomizing the data, the correct classification ranged between 89% and 100%, where most lay between 94% and 97%. The pairwise post-hoc Hotelling tests for the randomized subsets showed significant differences with all subsets of RW1 after sequential Bonferroni correction. The pairwise post-hoc Hotelling test showed significant differences between RW2 and WCTM. However, after sequential Bonferroni correction 70% was not significant. The pairwise post-hoc pairwise post-hoc Hotelling tests for randomized RW2 and WCTF showed significant difference in howls for all subsets. After sequential Bonferroni correction 80% was significant. The pairwise post-hoc Hotelling for randomized WCTM and WCTF showed significant differences for most subsets. After sequential Bonferroni correction 70% was not significant.

### 3.3. Individual Identification of Eurasian Wolves

Ninety-seven howls from nine Eurasian wolves were used in this analysis. Using eight variables (Minf0, Maxf0, Posmaxf0, Sd, Cofm, Startf0, Endf0, and Duration), the discriminant analysis achieved 89% accurate classification (Figure 4). SK1, and ULF achieved 100% correct classification. BLS0028 achieved a correct classification of 94%, BLS011 had an 85% correct classification. BLS026 achieved 83% correct classification and BLS010 achieved an 82% correct classification. The pairwise post-hoc Hotelling test showed significant differences between howls from all except BLS011 and BLS026 (Table 6). After sequential Bonferroni correction BLS026 and SK1 and BLS011 and SK1 were not significant. After randomizing the data, the correct classification ranged between 89% and 97%, where most were between 92% and 97%. The pairwise post-hoc Hotelling test revealed 57% significant differences and 20% was significant after sequential Bonferroni correction was applied.

## 4. Discussion

We found that it is possible to use howls to identify individual wolves within two subspecies (Arctic and Eurasian) with high accuracy. However, although the individual identification in this study had correct classification of 80–97%, noise from the surroundings made several howls unusable for extracting the fundamental frequency.

Wolves howl most actively during the months of July through October [33], whereas we recorded from late September through November and again from March to April. Future studies may benefit from collecting data in the period where wolves are more actively howling to gain larger datasets.

### 4.1. Identification of Subspecies

The present study showed significant difference between howls from all the subspecies. For the subspecies Arctic wolves and Eurasian wolves there was a correct classification of 78% for both. This is in agreement with other studies that shows howls differed across subspecies with classifications of 87% and 81% [32,41]. For the subspecies Northwestern wolves, the correct classification accuracy was only 64%. This classification could be due to most recordings being chorus howls, making individual identification difficult.

### 4.2. Identifying Individuals

It was possible to identify individual wolves within the subspecies Arctic, and Eurasian wolves based on their acoustic profile. Arctic wolves showed an overall correct classification of 95% and several wolves from both Arctic and Eurasian subspecies reached 100% correct classification. These results are in agreement with other studies showing that it is possible to identify individual wolves within subspecies based on their howls [27,36,40].

Root-Gutteridge et al. [27,40] included amplitude variables to improve individual identification, which increased their accurate classification to 100%. However, amplitudes are affected by ambient noise (wind, birds, etc.), which was the case with many of the howls recorded in this study and amplitude was therefore not usable for identification of wolves. To implement the use of amplitude, meticulous collection of data is necessary. By eliciting the howling, as achieved in several other studies [32,35,42], ambient noise can be avoided in recordings and analyzing variables extracted from amplitude in addition to fundamental frequency (f0) is feasible. In addition, the surrounding terrain and distances to the wolf affects the amplitude [27]. With low quality recordings and low-cost user-friendly software, it is also possible to identify wolves, whereas more costly software such as MATLAB, may limit the access for scientists as funding and knowledge of programming is necessary to use such software [55]. Further, different studies use different software [31,40,42,43], which makes comparison of data and replication difficult [55].

### 4.3. Future Use of Acoustic Monitoring

Efficient monitoring of wolves is necessary to fulfill obligations in relation to the EU Habitats Directive and to address public concerns about the increasing populations across Europe. Acoustic monitoring is an approach that is already used for identifying different species [17,18,19,20,21,56,57], and over the years researchers have monitored wolves acoustically as well [25,39] and might be used to identify if there are any new wolves in an area [39]. If we want to improve the continuous monitoring of wolves in countries, such as Denmark, where the species is returning to its historic habitat, it would be advantageous to start acoustic monitoring. If we can identify known wolves on their howls, we expect to be able to identify wild wolves on their howls.

Camera traps are used today in monitoring wolves [15,30,39] and both camera traps and acoustic recorders can offer a possibility for a permanent catalogue of sounds from the surroundings [30,57]. A combination of camera traps and acoustic devices could be beneficial for the research of wolves, as it can help enhance the survey area and improve detectability [30,57]. Furthermore, this combination of methods may help in understanding the interaction between wolves and humans [57]. Cameras can show the presence, type and intensity of human disturbance whereas the acoustic device can capture the noise and its intensity [57]. Having a combination of camera traps and acoustic devices can show whether wolves move away from areas with too much human activity or noise, as some animals change behavior or increase their nocturnality in areas with human activity [58]. With evolving of technologies, wolf howls recorded on mobile phones may also, in the future, be useful for monitoring wolves acoustically. This way it will be possible to involve citizens to help gain knowledge of their distribution and re-productive success.

## 5. Conclusions

The individual identification of the captive subspecies Arctic wolves and Eurasian wolves was accurate, and subspecies could be differentiated from each other. Our results suggest that acoustic monitoring and individual identification of wild wolves is possible. This could be a valuable noninvasive and cost-effective tool to complement monitoring based on genetics analyses, camera trapping, and other methods.

## Figures and Tables

**Figure 1 animals-12-00631-f001:**
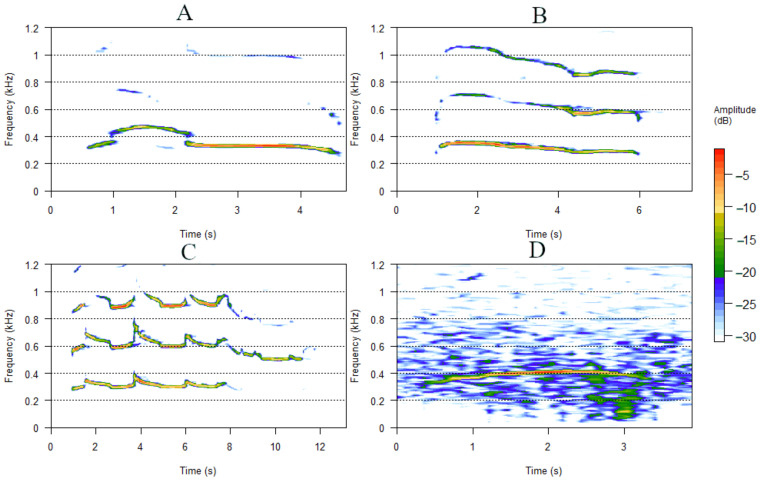
Example of spectrograms of single howls from (**A**). Northwestern wolves; (**B**). Arctic wolves; (**C**). Captive Eurasian wolves; and (**D**). Eurasian wolves with noise. The *y*-axis shows the frequency in kHz and the *x*-axis shows the time in seconds. Colors indicate the amplitude of the howls in dB (decibel). The lowest band in the spectrograms are the fundamental frequency (f0) and the other two are the second and third harmonic, which are visible in (**A**–**C**). The amplitude is in negative dB as 0 is referring to the maximum sound [51]. Graphs were made in the program RStudio [52] using the package Seewave [53].

**Figure 2 animals-12-00631-f002:**
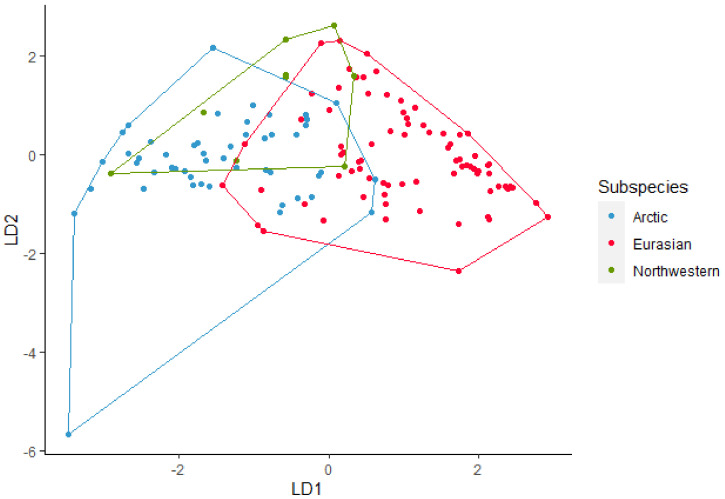
Linear discriminant (LD) analysis plot for identification of the subspecies Arctic, Eurasian, and Northwestern wolves with a 78% correct classification.

**Figure 3 animals-12-00631-f003:**
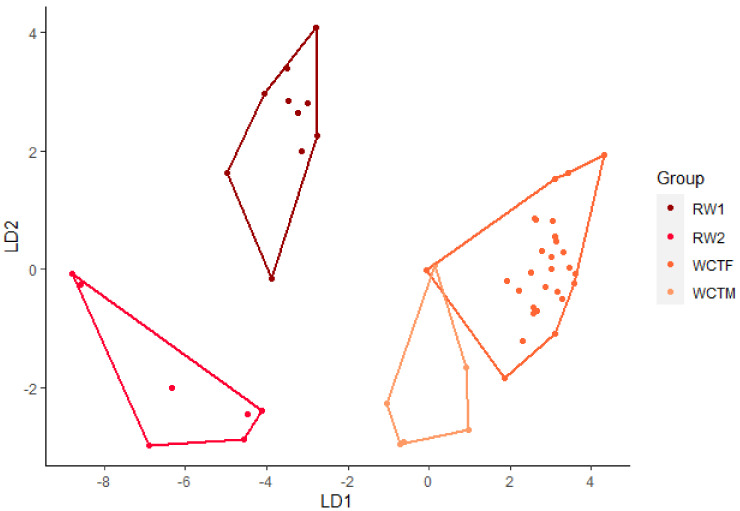
Linear discriminant (LD) analysis plot for individual identification of howls from Arctic wolves with a 95% correct classification.

**Figure 4 animals-12-00631-f004:**
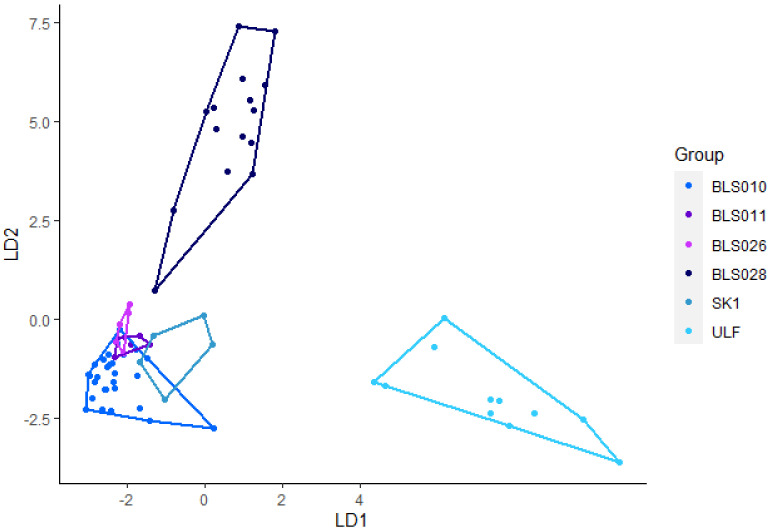
Linear discriminant (LD) analysis plot for individual identification of howls from Eurasian wolves with a 92% correct classification.

**Table 1 animals-12-00631-t001:** Wolves included in the individual identification with source for recordings, subspecies, and number of howls.

Wolf	Source	Subspecies	Scientific Name	Status	No. of Howls
GW1	Givskud Zoo, Denmark	Northwestern	*Canis lupus occidentalis*	Captive	2
GW2	Givskud Zoo, Denmark	Northwestern	*Canis lupus occidentalis*	Captive	7
GW3	Givskud Zoo, Denmark	Northwestern	*Canis lupus occidentalis*	Captive	2
SK1	Skandinavisk Dyrepark, Denmark	Eurasian	*Canis lupus lupus*	Captive	6
SK2	Skandinavisk Dyrepark, Denmark	Eurasian	*Canis lupus lupus*	Captive	5
Ulf	In the wild close to Ulfborg, Central Jutland, Denmark	Eurasian	*Canis lupus lupus*	Wild	13
BLS004	UK Wolf Conservation Trust	Eurasian	*Canis lupus lupus*	Captive	4
BLS010	British Library Sound Archive	Eurasian	*Canis lupus lupus*	Captive	34
BLS011	British Library Sound Archive	Eurasian	*Canis lupus lupus*	Captive	7
BLS026	Macaulay Sound Archive	Eurasian	*Canis lupus lupus*	Captive	6
BLS028	Poropets, Russia	Eurasian	*Canis lupus lupus*	Captive	18
BLS029	UK Wolf Conservation	Eurasian	*Canis lupus lupus*	Captive	4
WCTM	UK Wolf Conservation Trust	Arctic	*Canis lupus arctos*	Captive	7
WCTF	UK Wolf Conservation Trust	Arctic	*Canis lupus arctos*	Captive	35
RW1	Ree Park Safari, Denmark	Arctic	*Canis lupus arctos*	Captive	12
RW2	Ree Park Safari, Denmark	Arctic	*Canis lupus arctos*	Captive	8

**Table 2 animals-12-00631-t002:** Definition of acoustic variables assessed in wolf howls to investigate identification of subspecies and individual identification of wolves.

Variable	Definition	Extracted from
Meanf0 ^a,b,c^	Mean of the distribution of the fundamental frequency in Hz	Praat
Minf0 ^a,b,c^	Minimum of fundamental frequency in Hz	Praat
Posminf0 ^a^	Position of minimum frequency (time of min/duration)	Praat
Maxf0 ^a,b^	Maximum frequency in Hz	Praat
Posmaxf0 ^a^	Position of maximum frequency (time of max/duration)	Praat
Sd	Standard deviation of the mean fundamental frequency Hz	Praat
Cofv ^a,b^	Coefficient of frequency variation ((sd/Meanf0) × 100)	Praat
Slope ^c^	The mean absolute slope of the distribution of f0	MATLAB
Cofm ^a,b^	Coefficient of frequency modulation Σ|f(t) − f(t + 1)|/(n − 1)/Meanf0 × 100	MATLAB
Range ^b^	Range of the frequency (Maxf0 − Minf0)	Praat
Q25	25% of the distribution of the fundamental frequency in Hz	Praat
Q75	75% of the distribution of the fundamental frequency in Hz	Praat
IQR	Interquartile range of f0 (Q75 − Q25)	Praat
Startf0 ^c^	Start of the fundamental frequency in Hz	MATLAB
Endf0 ^a,b,c^	End of the fundamental frequency in Hz	MATLAB
Duration ^a,b,c^	Duration of the howl in seconds	MATLAB

Variables used in: ^a^ Hennelly et al., (2017) [32], ^b^ Root-Gutteridge et al., (2014) [27], ^c^ Watson et al., (2018) [50].

**Table 3 animals-12-00631-t003:** Removed variables and the variables they were correlated with from Pearson correlation test both for the analysis of Subspecies and for the analysis of individual identification within the two subspecies: Arctic and Eurasian wolves.

	Removed Variable	Correlated with
Subspecies	Meanf0	Maxf0, Endf0
	Range	Maxf0, Sd
	Posminf0	Posmaxf0
Arctic wolves	Range	Maxf0, Sd
	Posminf0	Posmaxf0
	Cofv	
	Slope	Endf0
	Range	Maxf0, Sd
	IQR	Startf0
Eurasian wolves	Meanf0	Minf0, Maxf0, Endf0
	Posminf0	Minf0
	Cofv	Maxf0
	Slope	
	Range	Startf0, Endf0
	IQR	Minf0

**Table 4 animals-12-00631-t004:** Pairwise post-hoc Hotelling test for Subspecies with F-values, *p*-values, and number of degrees of freedom (DF). Symbol ^#^ indicates significant *p*-values after sequential Bonferroni.

	F	*p*	DF
Arctic-Eurasian	23.77	<0.001 ^#^	9
Arctic-Northwestern	2.9	<0.01 ^#^	9
Eurasian-Northwestern	10	<0.001 ^#^	9

**Table 5 animals-12-00631-t005:** Pairwise post-hoc Hotelling test for Arctic wolves with F-values and *p*-values. Symbol ^#^ indicates significant *p*-values after sequential Bonferroni test.

	F	*p*	DF
RW1-RW2	33.2	<0.001 ^#^	8
RW1-WCTM	37.52	<0.001 ^#^	8
RW1-WTCF	93.87	<0.001 ^#^	8
RW2-WCTM	9.68	<0.01 ^#^	8
RW2-WCTF	41.74	<0.001 ^#^	8
WCTM-WCTF	8.2	<0.01 ^#^	8

**Table 6 animals-12-00631-t006:** Pairwise post-hoc Hotelling test for Arctic wolves with F-values, *p*-values and number of degrees of freedom (DF). Symbol ^#^ indicates significant *p*-values after sequential Bonferroni. n.s. indicates not significant *p*-values.

	F	*p*	DF
BLS010-BLS011	8.05	<0.001 ^#^	7
BLS010-BLS026	13.91	<0.001 ^#^	7
BLS010-BLS028	62.04	<0.001 ^#^	7
BLS010-SK1	58.31	<0.001 ^#^	7
BLS010-ULF	169.04	<0.001 ^#^	7
BLS011-BLS026	1.56	n.s.	7
BLS011-BLS028	9.21	<0.001 ^#^	7
BLS011-SK1	23.07	<0.01 ^#^	7
BLS011-ULF	32.85	<0.001 ^#^	7
BLS026-BLS028	10.2	<0.001 ^#^	7
BLS026-SK1	9.61	<0.05	7
BLS026-ULF	34.13	<0.001 ^#^	7
BLS028-SK1	17.44	<0.001 ^#^	7
BLS028-ULF	37.4	<0.001 ^#^	7
SK1-ULF	13.16	<0.001 ^#^	7

## Data Availability

The data presented in this study are available upon request from the corresponding author.

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
