# Peer review of "Bioacoustic Detection of Wolves: Identifying Subspecies and Individuals by Howls"

_animals, 2022, doi:10.3390/ani12050631_

Round 1

Reviewer 1 Report

This study analyses 170 howls of potentially 16 timber wolf callers of three subspecies for belonging to subspecies and individual. The topic is interesting and currently attracts a broad interest of researchers and conservationists. However, the MS should be thoroughly revised to meet the high standards of Animals.

Main concerns:

The authors should provide detailed description of how they identified particular individuals from the recordings made automatically and in archive recordings.

The authors use separate sets of call variables for each individual and for each subspecies. Moreover, sets of parameters may vary even for the same individuals or subspecies depending on analysis and sample of calls and animals. This approach makes their results incomparable between numerous own analyses as well as with other studies. The author should keep the set of acoustic parameters the same for all individuals or subspecies included in the given analysis.

The authors should provide the reasoning why they make so many similar analyses. Much better is to select one for individuality and one for subspecies.

Applied statistical tests make doubtful the adequacy of the obtained results and should be (where possible), replaced or corrected. The authors make multiple pairwise comparisons instead of using the statistics with post-hoc tests (or at least Bonferroni correction). Some individual wolves are represented with call samples of only 2 howls, what does not meet the demands of the used by authors discriminant analysis (DFA). Moreover, samples from different individuals in DFA are not equal. This makes questionary both applicability of DFA in principle for samples of this study and reliability of the obtained results. At least, for each DFA, it is necessary to correct the obtained result, by calculating the random (by chance) value and to compare it with the observed value.

Writing of MS should be substantially improved. The problem is not with English, but with very poor structure of the text. This makes Methods unclear for the reader and thus study is non-reproducible.

Abstract

L 18 from the three subspecies

Add “:” after subspecies

L 29 lowest frequency of a sound signal

Add “band” after frequency

L 36 Keywords: habitat directive

What is habitat directive? Is this usual term?

Introduction

L 43-44 In Europe (excluding Russia), there are more than 17,000 wolves [2].

Why excluding Russia, as a part of work was done in Russian facilities?

I looked Internet and found immediately some references, providing some information regarding Russian wolf populations:

Korablev, M.P., Korablev, N.P. & Korablev, P.N. Genetic diversity and population structure of the grey wolf (Canis lupus Linnaeus, 1758) and evidence of wolf × dog hybridisation in the centre of European Russia. Mamm Biol 101, 91–104 (2021). https://doi.org/10.1007/s42991-020-00074-2

Aspi, J., Roininen, E., Kiiskilä, J. et al. Genetic structure of the northwestern Russian wolf populations and gene flow between Russia and Finland. Conserv Genet 10, 815–826 (2009). https://doi.org/10.1007/s10592-008-9642-x

L 70-71 Acoustic monitoring is a passive monitoring tool that has been used in the last decades for studies of diverse taxa, including insects [15], bats [16,17], birds [18], and whales 71 [19,20],…

Acoustic passive monitoring was also applied for monitoring large terrestrial mammals, as e.g., ungulates:

Rusin I.Y., Volodin I.A., Sitnikova E.F., Litvinov M.N., Andronova R.S., Volodina E.V. Roaring dynamics in rutting male red deer Cervus elaphus from five Russian populations. Russian Journal of Theriology, 2021, V. 20, N 1, P. 44–58. doi: 10.15298/rusjtheriol.20.1.06

Volodina E.V., Volodin I.A., Frey R. Male impala (Aepyceros melampus) vocal activity throughout the rutting period in Namibia: daily and hourly patterns. African Journal of Ecology, 2021. doi: 10.1111/aje.12923

L 54-55 wolves are both wide-ranging [5]

Another relevant reference: Kirilyuk et al. 2020: Long-distance dispersal of wolves in the Dauria ecoregion. Mammal Research 65:639-646

L 77-78 However, most of studies of acoustic have been based on North American or Indian wolf populations and there is only limited evidence for its potential in Europe.

This is incorrect. There are many papers on the acoustics of wolf packs in Europe. See e.g.:

Palacios V., Font E., MaÒ‘rquez R., 2007. Iberian wolf howls: acoustic structure, individual variation, and a comparison with North American populations. J Mammal., 2007, v. 88, p. 606–613. https://doi.org/10.1644/06-MAMM-A-151R1.1

Mazzini F., Townsend S.W., Virányi Z., Range F., 2013. Wolf howling is mediated by relationship quality rather than underlying emotional stress. Curr. Biol., 2013, V. 23, N 17, p. 1677–1680. doi: 10.1016/j.cub.2013.06.066

Watson S.K., Townsend S.W., Range F., 2018. Wolf howls encode both sender- and context-specific information. Animal Behaviour, 2018, v. 145, p. 59-66. doi. 10.1016/j.anbehav.2018.09.005

Kershenbaum A., Deaux E.C., Habib B., Mitchell B., Palacios V., Root-Gutteridge H., Waller S., 2017. Measuring acoustic complexity in continuously varying signals: how complex is a wolf howl? Bioacoustics, 2018, v. 27, N 3, p. 215-229. doi. 10.1080/09524622.2017.1317287

Passilongo D., Buccianti A., Dessi-Fulgheri F., Gazzola A., Zaccaroni M., Apollonio M., 2010. The acoustic structure of wolf howls in some eastern Tuscany (central Italy) free ranging packs. Bioacoustics, 2010, V. 19, N 3, P. 159-175.

L 87-88 …the lowest frequency of a sound signal and is perceived as the sound’s pitch.

change to “the lowest frequency band of the tonal call as wolf howl.”

L 87-88 …the lowest frequency of a sound signal and is perceived as the sound’s pitch.

Delete “and is perceived as the sound’s pitch”. People would perceive f0 even if it is cut off at all. For example, in old stationary telephones, the f0 of human speech was cut off perfectly, however people could talk and understand the speech without problems. Human brain re-constructs the f0 on the basis of ratios of the upper harmonics. In wolves, f0 is very close to those of human speech so people can perceive f0 similarly to human speech.

L 91-93 Root-Gutteridge et al. [36] were able to identify individuals in captive Eastern wolves with 100% accuracy using both f0 and amplitude (the highest variation of a wave in air pressure, perceived as volume).

I think that the authors mean “verifying” rather than “identifying” but in this case. In captivity, investigators know the actual number of individual wolves in the study pack and only verify their identity by their howls. This is distinctive from the situation in the field, where researchers do not know a priory the number of howling animals. In this case, the task of identifying the animals by their howls is much more complicated. And it is different from captivity, where the animals are in the same small enclosure and therefore at approximately the same distance from the researcher. In the field, individuals can be dispersed so howl amplitude will be not so reliable key to individual identity as in captivity.

L 99-100 Efficient monitoring of wolves is necessary to fulfill obligations in relation to the EU Habitats Directive and to address public concern about the increasing populations across Europe.

If the authors want to keep this statement in Introduction, they should explain somehow why they only restrict applicability of their research within EU. Is it only actual within this region? As well as I know, Animals is an international journal, publishing basic rather than applied research. Otherwise, I suggest transferring this statement to Discussion, where it will be well appropriate, with adding references to relevant EU documents.

L 101-102 The monitoring based on camera traps requires individuals in near field of view whereas acoustic recorders cover larger distances.

Please indicate the distances covered with camera traps and by acoustic monitoring with relevant references.

L 114-115. Furthermore, we aim to discuss the usefulness of acoustic monitoring in the field as only limited data are available from Europe.

Please re-phrase this sentence because data from Europe are indeed extensive (see comment to Lines 77-78).

Materials and Methods

L 125 Full details are given in Table 1.

Change Table 2 and Table 1. Table 1 describes acoustic parameters measured in other papers but does not provide any information about study locations. In Table 2 (indeed Table 1) the authors should better describe each location and indicate whether the wolves were captive or free-ranging, for each location.

Table 2

Ulfborg – what location is it? A description is lacking in the text.

Change Holly Root-Gutteridge to Toropets, Russia

L 128-130 The acoustic recorder had a sampling rate (average number of samples per seconds of an audio recording) of 44.1 kHz and an amplitude resolution of 16 bits (number of amplitude values possible to record for each sample).

Delete average before number. Change number to digit capacity.

L 135 UHER 4000 REPORT-L

This is tape recorder, not a digital recorder.

L 136-137 Marantz PMD recorder

PMD-660, PMD-670, PMD-661, PMD-671 or another?

Please indicate the firm producer, country and headquarters town after UHER, Marantz and Sennheiser (as you did it after Song Meter)

L 143 Sennheiser ME67 directional long-range microphone, K6 power unit,

Change to Sennheiser K6-ME67

Please indicate the firm producer, country and headquarters town after M-Audio Microtrack, Sanyo.

L 146-150 Howls obtained from British Library Sound Archive were recorded by Dr. Erik Zimen in the 1970s [39] reported in the notes. Howls obtained from Macauley Sound Archive were recorded with UHER 4000 RE-PORT L recorder using a Sennheiser MKH104 condenser microphone reported in the notes.

Please indicate what equipment was used for digitizing these tape recordings? At what sampling rate and resolution?

L 151-156 In Givskud Zoo a portable sound amplifier (Joyo JPA-863) was used to elicit howling. The howls were initiated by playing sounds for the wolves. Once they started howling the sound was stopped. When the wolves had calmed down and they no longer barked, howled, or whined another sound was played for them. This was continued for an hour with alternating sounds of ambulance sirens, church bells and howls from a different wolf pack.

Replace this paragraph to the description of recordings in Denmark. Please indicate firm producer after Joyo.

L 151 All recordings were saved in the file format wav.

Please indicate, at what sampling rate and resolution. If they are the same for all recordings, please indicate this one time. If no, then indicate for each recording. It is important, because the sampling rate value directly influences the resolution of spectrographic analysis.

L 157-159 Sound signals are made up of partials [40] also known as harmonics [41] (Figure 1). The fundamental frequency (f0) is usually the lowest frequency of harmonics in a sound signal.

Delete this. You correctly write further that f0 corresponds to the rate of vocal folds vibration. Harmonics represent integer multiples of f0 band; they arise at constructing call FFT spectrum, because sound wave created in sound source (the glottice) differs from sinusoid. The f0 band is not a harmonic.

L 175 2.2. Software for identification of howls

Consider changing the title to “Call analysis”. This is much more relevant to the content of this section.

L 190-194 For the data collected in Denmark, Arctic and Eurasian wolves were identified on 190 their solo howls. Some of the wolves would howl unaccompanied for several minutes and were identified as one individual. These were compared to other solo howls from the same location to determine whether they were from another individual, or they were from the same individual. Two of the wolves had very distinct vocalizations in their howls. Howling Northwestern wolves were filmed and individually identified.

Based on this unclear explanation, the reader cannot reproduce this study. Please re-write to explain better how you did the choice of calls, how compared then and how assigned calls to particular individuals.

L 196 the pitch

the pitch=f0? If yes, delete the pitch. If no, please explain what do you mean under the pitch. Please avoid synonyms throughout the text. And please explain how the pitch was tracked in the calls?

L 198-200 Nineteen acoustic variables (Table 1) ...; where 13 variables have been used in other studies

Change Table 1 to Table 2.

There are only 17 acoustic variables in Table, and only 12 of them have been used in other studies

L 200 The Vocal parameters were measured

Please indicate sampling rate and resolution for the wav-files included in analysis. You should have made them the same for all recordings, as the value of sampling rate directly influences resolution of spectrographic analysis.

L 201-202 using a cross-correlation method (Sound: To 201 Pitch (cc) time step of 0.005 s, pitch floor 75 Hz, pitch ceiling 1200 Hz)

These settings are for PRAAT, not for MATLAB. Remains unclear, did the authors use for measuring f0 parameters PRAAT or script in MATLAB?

Please provide references or indicate firm producers for MATLAB and PRAAT.

Table 1

IQR Interquartile range (25%-75%)

Do you mean Interquartile range of f0 values (but in this case, is unclear, how it was calculated, indicate this in Table or in the text). Or do you mean the values of call power spectrum (but in this case, this parameter is not related to f0)?

Skewness Skewness of the distribution of the fundamental frequency.

Please indicate how this parameter was calculated, to make your data reproducible.

Kurtosis Kurtosis of the distribution of the fundamental frequency

Please indicate how this parameter was calculated, to make your data reproducible.

Please indicate the units for Slope, Range, IQR, Skewness and Kurtosis.

L 209-210 A total of 265 solo howls from 11 wolves were recorded during the study period in 2020-2021.

Earlier you wrote that Data collections took place between July 2011 and April 2021 (L 188). Or this is some dataset unknown for the reader? So, please indicate what recordings are in this dataset.

How did you know that all 11 individuals provided the howls, if only 55 howls could be assigned to 7 individual wolves (L 211-212)?

L 213-214 In Ree Park Safari and Skandinavisk Dyrepark we identified 2 out of 3 wolves. In Givskud Zoo we identified 3 out of 7 adult wolves.

Please describe in detail, how you identified particular individuals from the recordings made automatically (L 127).

L 214 Each of the 11 wolves

2+2+3=7. See Table 2(1)

L 216 We also used archival howls

Did these archive recordings also represent solo howls? If no, please write how you separated calls from one individual wolf from another on the archive recordings (for example, male and female)? Also, in case if these are not solo howls, could some differences between howls outcome from the way how the howls are produced, solo or in a group?

L 217-219 50 wolf howls collected from The UK Wolf Conservation Trust and 18 Eurasian wolf howls collected in Russia by HRG.

Were these recordings archiving recordings? In the text, is indicated that at least some of them were collected by one of the authors (HRG). Separation of recordings to the collected and archive recordings is confusing for the readers.

L 240-242 A MANOVA was used to test if there was a statistically significant difference between the howls in subspecies.

Please indicate which parameters you compare using MANOVA. The Canonical scores of discriminant analysis or parameters included in DFA or all acoustic variables?

You make multiple pairwise comparisons (here and further in the text). Why you do not apply correction (for example, Bonferroni correction) or do not apply statistics with post-hoc tests?

L 256-260 A discriminant analysis for Arctic wolves was applied to 10 of the 17 variables for the Arctic wolves (Meanf0, Minf0, Maxf0, Posmaxf0, Sd, Cofm, Startf0, Endf0, IQR, and Duration), 10 of the 17 variables for Eurasian wolves (Meanf0, Posmaxf0, Sd, Cofv, Slope, Cofm, Range, Startf0, Endf0, and Duration), and nine of the 17 variables for Northwestern wolves (Minf0, Maxf0, Posmaxf0, Slope, Cofm, IQR, Startf0, Endf0, and Duration).

On me, different variables for each subspecies make results incomparable between subspecies. Why not using the same small set of variables equally appropriate for each subspecies?

Some individual wolves are represented with call samples of only 2 howls. On my opinion, this is very little for using discriminant analysis. Moreover, samples from different individuals in DFA are not equal. This makes questionary both applicability of DFA in principle for samples of this study and reliability of the obtained results. At least, for each DFA, it is necessary to correct the obtained result, by calculating the random (by chance) value and to compare it with the observed value.

Effects of different number of samples per group for DFA results and possible ways of correction:

Titus K., Mosher J.A., Williams B.K., 1984. Chance-corrected classification for use in discriminant analysis: ecological applications. American Midland Naturalist, v. 111, p. 1-7.

Calculating DFA random value:

Solow A.R., 1990. A randomization test for misclassification probability in discriminant analysis. Ecology, v. 71, p. 2379-2382.

Applying Solow, 1990, example:

Volodin I.A., Volodina E.V., Frey R., Karaseva K.D., Kirilyuk V.E. Daurian pika (Ochotona dauurica) alarm calls: individual acoustic variation in a lagomorph with audible through ultrasonic vocalizations. Journal of Mammalogy, 2021, V. 102, N 3, p. 947-959. doi: 10.1093/jmammal/gyab048

Another approach for correction of DFA and calculating the random value:

Mundry R., Sommer C., 2007. Discriminant function analysis with nonindependent data: consequences and an alternative. Animal Behaviour, v. 74, p. 965-976. doi:10.1016/j.anbehav.2006.12.028

L 265-266 This was done by doing the MANOVA on two individuals at a time.

You make multiple pairwise comparisons. Why not using one analysis with post-hoc test?

Table 4 Removed variables and the variables with which they were correlated from the Pearson correlation test for Eurasian wolves.

Deleting different numbers of variables from each particular analysis makes the obtained results incomparable.

L 290 2.6. Individual identification of wolves from two subspecies on their howls

On my opinion, this is redundant analysis, which does not provide any new information to the comparison of subspecies and individual based variation of wolf howls. If you consider that it is necessary, please provide the reasons for it.

Results

L 321 F = 3.3

Here and in the remaining text. Please indicate the numbers of degrees of freedom for F-ratios.

Figure 3.

This figure is incorrect and misleading, as GW1 and GW3 are represented with only two points and cannot create figures in the space of two first axis of DFA.

Figure 6.

Similarly, as GW1 and GW3 are represented with only two points, they should be presented on the figure in the form of line.

Discussion

L 410-412 We found that it is possible to use howls to identify two subspecies of wolves (Arctic and Eurasian), ... with high accuracy.

This conclusion is incorrect, please delete it. DFA results cannot be interpreted partially, only for two groups of the three included in analysis. As these results depend directly on the number of groups, number of samples and number of parameters included in analysis. So, comparison of DFA results is only possible if all these numbers are equal in different analyses.

L 412-414 However, although the individual identification in this study had correct classification of 90-100%, noise from the surroundings made several howls unusable for individual identification.

If you extracted f0 and measured its parameters, the surrounding noise could not interfere the identifying.

L 420-421 The present study showed significant difference between howls from all the subspecies.

For this conclusion, the authors should make a correct DFA and to calculate the random value of correct assignment.

L 427-430 It was possible to identify individual wolves within the subspecies Northwestern, Arctic, and Eurasian wolves based on their acoustic profile. Northwestern wolves showed an overall correct classification of 100% and several wolves from the Arctic and Eurasian subspecies reached 100% correct classification.

Similarly, for such conclusion, the authors should make a correct DFA and to calculate the random value of correct assignment. Also, DFA results cannot be interpreted partially only for some individualsincluded in analysis (as discriminant keys are calculated on the basis of all samples=number of howls here).

L 457-458 The larger number could increase the similarity between the parameters extracted from the howls.

I do not understand how the increase of the number of animals can result in increase the similarity between the acoustic parameters. Please explain or delete.

L 462-464 However, the Arctic and Eurasian wolves seem to cluster based on location rather than subspecies. This indicates that group specific signatures in wolf howls are more prevalent than subspecies specific signatures.

It is unclear from the text, on what this conclusion is based. Please explain.

Author Response

We thank you for your comments to our manuscript.

Sincerely,

Hanne Lyngholm Larsen

Reviewer 2 Report

Topic of the manuscript is strongly interesting and underline acousting monitoring of protected species as one among other methods. Methodology, as well as results are well documented, with details, there is no lack of anything. Detailed information, especially in Chapter 2: Material and Methods, are important to understand methodology properly.

I have minor suggestions of changes, e.g.:

  1. Title. I would make it a little shorter:

Bioacoustic Detection of Wolves: Identifying Subspecies and Individuals Wolves by Their Howls. 

It is logical, so we do not have to repeat "wolves" word, as well as their - nothing else. 

2. Line 24 & 25:  We analyzed acoustic data of wolves’ howls collected from both wild and captive wolves.  I would write:

We analyzed acoustic data of wolves’ howls collected from both wild and captive ones.

3. Line 25 - without coma. 

4.  Lines 18 & 19 with comparing to 27-28. In lines 18&19 I would write full Latin name only with the first species name - later C.l. the same like in Abstract.

5. Line 68&69 - try to thick your text. Don't leave "area" alone in one line.

6.  165.  Prepare signatures under the table in the same way, I mean:  aVariables used in Hennely et al., (2016). bvariables used in Root-Gutteridge et al., (2014), cused in Watson et al., (2018). 

Dots or comas between a and b, b and c.

Repeat Variables everywhere in a, b and c, or write: Variables used in: axxxx, bxxxxxx, cxxxxxxx.

7.  Line 200 - vocal, not Vocal.

8. Lines 385, 386 and 394: there is no need of dots at the end of titles.

9.  Line 474: A combination, not A Combination

10. Line 484 - try to thick your text.

11. Line 510: references:  use this bullet style:

  1. xxxxxxxxxxxxxxxxxxxxxxxxxxxxxxxxxxxxxxxxxxxxxxxxxxxxxxxxxxxxxxxxx.
  2. xxxxxxxxxxxxxxxxxxxxxxxxxxxxxxxxxxxxxxxxxxxxxxxxxxxxxxxxxxxxxxxxx.

12. Lines 512, 519, 525, 531, 534 - delete ";," at the ends.

13. Lines 525, 556,573, 577, 587, 591, 604, 616, 620, 625 - according to MDPI requirements you should not put "" but shorter one "-".

14. 536 - lack of pages

Author Response

We are thankful for your comments to our manuscript.

Sincerely,

Hanne

Reviewer 3 Report

Comments on MS animals-1590081

This is an interesting study about identifying wolves and wolf populations based on howl. The results are clearly presented and the sample sizes are adequate despite the problems with the recordings. I’m not an expert in bioacoustics, so I can’t comment on the method per se.  The statistical methods used are appropriate. 

The authors suggest that the method is suitable for monitoring the sizes of wolf populations. The method is probably best suited to countries where the number of wolves are low or where they are easy to trace due to e.g. the small territory size of the wolves.  Acoustic-based methods are not very cost-effective in countries where the wolf populations are sparse and the territories are large.  In these countries, telemetry or DNA methods are more effective tools in assessing population sizes. 

Author Response

We are thankful for your comment. 

We think that with time the wolf population in Denmark is expected to increase the method will be expected to be useful in future monitoring of wolves in Denmark and other Western European countries.

Sincerely,

Hanne Lyngholm Larsen

Round 2

Reviewer 1 Report

In this revised version, the authors addressed all my comments. I recommend accepting the manuscript with minor corrections.

L 87 Change C.l.signatus to C.l. signatus

L 95-96 acoustic recorders cover larger distances with a diameter of 28 km [28].

This is impossible. Suter et al. 2017 [28] write about the total square of 28 km2, where their five automated devices recorded the howls of wolves. Please correct this.

L 229 2.3. Identification of Subspecies

Change to 2.4

L 250 2.4

Change to 2.5

L 268 Table 3

Add Arctic wolves to Table 3

Author Response

Dear reviewer,

Thank you for your comments. Answers have been added in italic below

L 87 Change C.l.signatus to C.l. signatus –  This has been changed.

L 95-96 acoustic recorders cover larger distances with a diameter of 28 km [28].

This is impossible. Suter et al. 2017 [28] write about the total square of 28 km2, where their five automated devices recorded the howls of wolves. Please correct this. – This has been changed to “with a radius of 3 km”

L 229 2.3. Identification of Subspecies

Change to 2.4 – This has been changed

L 250 2.4

Change to 2.5 – This has been changed

L 268 Table 3

Add Arctic wolves to Table 3 – Arctic wolves are already part of Table 3. However, subspecies have been added to the table. I hope that is what you meant.

Sincerely,

Hanne Lyngholm Larsen